# The relationship between auditory processing disorder and articulation disorders

**Safa Alqudah**[1,2]*, **Lama Al-rashed**[3], **Lelas Mansi**[3]

**1** Department of Rehabilitation Sciences, Faculty of Applied Medical Sciences, Jordan University of Science and Technology, Irbid, Jordan, **2** Speech-Language Pathology and Audiology Department, College of Medical Rehabilitation Sciences, Taibah University, Medina, Saudi Arabia, **3** Jordan University of Science and Technology, Irbid, Jordan

* Salqudah@just.edu.jo

## Abstract

### Purpose

Auditory processing disorder (APD) may attribute to certain speech problems, as auditory processing plays a vital role in phonetic development during childhood. This research investigates the incidence of APD in individuals with misarticulations of the /r/, /s/, /k/, /θ/, /dʒ/, and /q/ speech sounds. These sounds represent different places of articulation, manners of articulation, and phonation.

### Method

A total of 51 participants with articulation disorders aged 6–18 years underwent conventional peripheral assessments, including otoscopic examination, immittance measurement, and audiometry. Central auditory processing was also evaluated using a battery developed by Frank Musiek.

### Results

Central auditory processing evaluations revealed abnormalities in 37 of 51 participants (72.55%). Correlations between various sound misarticulations and APD were identified: 69.5% for /r/, 40% for /s/, 80% for /k/, 100% for /θ/ and /q/, and 83.3% for /dʒ/.

### Conclusion

The correlation between articulation disorders and APD varies depending on the specific sound affected. Further research is needed to establish clearer associations between different articulation disorders and APD. These findings underscore the importance of assessing central auditory system function in patients with articulation disorders before initiating speech therapy.

**Data availability statement:** All relevant data are within the manuscript and its Supporting information files.

**Funding:** This research was funded by the Jordan University of Science and Technology (Grant No. 20230618). The funder had no enrollment in the study design, data collection and analysis, decision to publish, or manuscript preparation.

**Competing interests:** The authors have declared that no competing interests exist.

## 1. Introduction

Auditory processing disorder (APD) is characterized by hearing difficulties resulting from deficits in the central auditory nervous system [1]. APD may affect one or more auditory skills, including temporal masking, sound localization, lateralization, auditory discrimination, temporal integration, temporal resolution, and temporal gap detection [2]. According to the American Speech-Language-Hearing Association [3], most individuals with APD report trouble understanding speech, particularly in noisy environments, poor short-term memory, and challenges in learning or retaining orally presented information. Additionally, they may also struggle to follow verbal directions, comprehend and recognize rapid or degraded speech, develop musical abilities such as recognizing sound patterns or rhythms, and may present with problems in receptive and expressive language skills [4]. The symptoms of pediatric APD overlap with those of the other common behavioral and intellectual pediatric conditions, including learning disabilities and attention deficit hyperactivity disorder [1]. APD can coexist with other childhood cognitive disorders because the central nervous system is nonmodular, meaning dysfunction in one region can impair processing in another [5]. Moreover, various perspectives have been proposed regarding the dimensions of APD. Inconsistencies in the literature arise from the multiple definitions of auditory processing, the lack of a standardized reference for APD diagnosis, and the diverse symptoms reported by patients. Consequently, various treatment approaches have been developed for APD management; however, the relationship between auditory processing deficits and language and speech impairments, as well as optimal strategies to address them concurrently, remains unclear [2,6,7]. Given the complexity of auditory processing, some scholars recommend a comprehensive APD assessment to determine specific disability rather than offering a general diagnosis. For example, a defect in temporal processing is associated with auditory discrimination difficulties and may be described specifically rather than broadly categorized as APD [8].

Language development is closely linked to auditory perceptual abilities, particularly pre-literacy skills, reading, and comprehension. Over time, auditory and verbal abilities become essential for academic success [9]. Language processing is complex, involving auditory, cognitive, and verbal components [2,10]. One study found that the rapid processing of acoustic information is crucial for linguistic development and maintenance [11]. Children with speech defects differ significantly from typically developing peers in terms of sensory information access speed. Speech intelligibility may also predict the ability to process fundamental sound parameters, such as duration and frequency [12]. Discriminating changes in frequency are necessary to distinguish between sound productions based on place and manner of articulation [13]. The acoustic cue for sound placement is the formant transition, which involves rapid frequency modification over a brief time interval [13]. Additionally, changes in duration likely influence the recognition of voice contrasts in sounds, differentiating between the silent bursts of stop consonants and turbulent airflow of fricatives [14]. Thus, there is a relationship between reduced ability to perceive subtle timing differences in continuous sound production and errors in voice recognition, even when hearing thresholds are within normal limits. For example, one temporal discrimination

required during fluent speech is discriminating the difference between the duration of voice onset time for consonants such as /p/ versus /b/, a difference of approximately 20 milliseconds [15]. Another study observed a link between diminished audio-temporal processing ability and speech disorders [16]. Reports indicate that children with deficits in time-processing skills struggle to detect phonetic changes in speech production [16]. Even recognizing word stress patterns involves identifying phonetic changes over time [17]. APD influences the severity of speech disorders in children, with more severe speech disorders associated with a higher likelihood of APD [18]. This area of research warrants further investigation, as current studies primarily define a general relationship between APD and sound misarticulation rather than pairing specific auditory processing skills with particular sound productions. Additionally, few studies have explored the relationship between APD and articulation disorder.

This study focused on identifying the incidence of APD in patients with articulation disorders involving abnormal production of the /r/, /s/, /k/, /θ/, /dʒ/, and /q/ sounds. The outcomes have significant implications for speech therapy by supporting the diagnosis of central auditory dysfunction related to sound production deficits. As a result, more effective treatments can be developed for patients with articulation disorders, advancing beyond traditional methods. Additionally, for patients with cochlear implants or hearing aids, these findings can facilitate specialized treatment for APD, helping them develop the auditory skills necessary to achieve target sound productions and speech outcomes more efficiently. In summary, this research provides valuable insights for speech therapy, improving diagnostic accuracy and treatment strategies for various challenges.

## 2. Methods

### 2.1. Study subjects

A cross-sectional study was conducted with 51 participants aged 6–18 years who had articulation disorders. The study was conducted with a 95% confidence interval, 80% statistical power, and a 5% margin of error. Evaluations followed the guidelines outlined by Dr. Frank Musiek and were conducted at King Abdullah University Hospital in Irbid, Jordan, from November 2, 2023, to August 31, 2024. Based on the criteria of [1,2], participants were diagnosed as having either normal auditory processing or APD. The inclusion criteria were as follows: (a) Normal hearing sensitivity across all tested frequencies; (b) normal middle ear function (type A tympanometry); (c) completion of APD subtests included in Dr. Frank Musiek's battery; and (d) complaints limited to misarticulations (pure articulation disorders based on SLP evaluations) of the /r/, /s/, /k/, /θ/, /dʒ/, and /q/ sounds resulting from idiopathic speech sound disorder (SSD). These errors were either distortions or single-sound substitutions that did not follow a rule-based phonological pattern, supporting their classification as articulatory in nature. This selection ensured a homogenous sample of children with functional SSD, allowing for a focused investigation of auditory processing abilities without interference from broader phonological or neurological conditions.

The exclusion criteria were as follows: (a) Participants diagnosed with peripheral hearing loss of any type and degree; (b) participants with speech and language disorders other than articulation disorders, whether isolated or comorbid; (c) participants with mental or physical dysfunction that could affect speech production; and (d) participants with speech muscle weakness or childhood apraxia of speech.

### 2.2. Test battery for APD

The auditory processing battery used in this study, compiled by Dr. Frank Musiek, evaluated APD through the Gaps in Noise (GIN) test, Duration Pattern Sequence (DPS) test, Pitch Pattern Sequence (PPS) test, Masking Level Difference (MLD) test, and Random Gap Detection Test (RGDT). The normative data for these tests are as follows: RGDT and MLD are applicable for individuals aged 5 and older, and GIN, PPS, and DPS for those aged 7 and older [19–23]. All APD tests were administered at 55 dB HL [19–23].

The GIN test measures temporal resolution and the ability to detect rapid changes in auditory stimuli. The MLD test at 500 Hz evaluates the binaural interaction mechanism of auditory processing. The PPS test assesses the ability to identify patterns of high- and low-frequency tones in the correct order of presentation. This test consists of three tones presented to the listener, who must describe each sound in the same sequence. It is suitable for individuals aged seven years and older and administered to each ear individually. Although the test consists of 60 patterns, it can be terminated if the subject correctly identifies the first 15 patterns, indicating typical temporal sequencing and pattern perception. The DPS test evaluates the listener's ability to detect changes in the duration of stimuli. It is conducted by presenting three tones, differing in duration, to each ear individually. The tones, either long or short, are presented in various sequences, and subjects are asked to describe them in the same order they heard. As with the PPS test, participants were presented with 60 patterns unless they correctly answered the first 15, in which the results were considered normal, indicating no abnormality in temporal ordering or pattern perception. The RGDT rigorously assesses an individual's ability to detect and identify brief gaps in auditory stimuli, reflecting the temporal complexities involved in speech sound discrimination. Participants are exposed to broadband noise bursts containing inserted gaps, with durations ranging from 2 to 20 milliseconds, and are randomized between 500 and 1000 milliseconds. They are instructed to listen carefully and indicate when they perceive a gap. The RGDT detects dysfunction in the timing processing of the auditory system, known as temporal processing disorder, in both children and adults [24]. Children with temporal processing disorder may struggle to develop the phonological skills necessary for efficient and automatic decoding and encoding of words while reading and writing. These skills are essential for mapping phonemes to graphemes [24]. Participants received repeated instructions to ensure they understood how to complete the tests. Tests were skipped if participants did not understand the instructions or gave questionable responses.

### 2.3. Auditory tests and speech assessments

A qualified speech-language pathologist evaluated participants' speech through a comprehensive assessment, including standardized and non-standardized language assessments, hearing evaluation, orofacial examination, Amayreh's articulation test [25], and spectrogram analysis. All participants were required to have a diagnosis of articulation disorder without comorbidity; those diagnosed with other speech disorders were excluded.

Auditory and speech assessments were conducted at King Abdullah University Hospital. The research team adhered to safety and hygiene protocols. The auditory diagnostic battery was administered following guidelines from the American Speech-Language-Hearing Association and the American Academy of Audiology [1,2]. Parents or guardians signed informed consent forms. The procedures complied with research protocols for human subjects and were approved by the Institutional Research Board of the Jordan University of Science and Technology (2/164/2023).

### 2.4. Statistical analysis

Statistical analyses were performed using Statistical Package for the Social Sciences software (version 21.0). Continuous variables are presented as mean±standard deviation (SD), while categorical variables are summarized using frequencies and percentages. Fisher's exact test was applied to assess the association of GIN, MLD, RGDT, and APD with sound mispronunciations. Repeated measures analysis of variance (ANOVA) within-subjects factors (Ear) was used to examine differences in PPS and DPS test scores related to sound mispronunciations. Linear regression quantified the impact of sound mispronunciations on PPS and DPS test outcomes. A significance level of 0.05 was set to determine statistical significance.

## 3. Results

### 3.1. Demographic characteristics of the study subjects

A total of 51 participants (age range: 6–18 years) diagnosed with pure articulation disorders were recruited. Speech and language impairments are more common in males than females, as reflected in the sample's distribution (70% males

versus 30% females). The majority of participants were elementary school students (72%). For clarity, educational levels were categorized as follows: elementary (ages 6–10 years), middle school (ages 11–14 years), and high school (ages 15–18 years). Table 1 presents the demographic details of the participants.

Participants exhibited two primary types of articulation errors: substitution (n = 28) and distortion (n = 23). Among those with substitution errors, 20 participants (71.4%) had APD, while 8 (28.6%) demonstrated normal auditory processing. Similarly, among participants with distortion errors, 17 (73.9%) had APD, and 6 (26.1%) exhibited normal auditory processing. Overall, 37 of 51 participants (72.5%) were diagnosed with APD, regardless of the type of articulation error, highlighting a strong association between articulation difficulties and auditory processing deficits. Notably, all participants exhibited articulation disorders rather than phonological disorders, as their errors were limited to distortion and substitution rather than phonological rule-based processes.

The Percent Consonants Correct–Revised (PCC-R) metric was used to assess the severity of speech sound errors. PCC-R scores among participants ranged from 76% to 97%, indicating varying degrees of mild to moderate SSD severity. According to Shriberg et al., PCC-R values above 85% indicate a mild disorder, while scores below this threshold suggest moderate impairment [26].

The PCC-R and distortion rates varied depending on the specific sound, with /r/ at 86.5% and 24%, /s/ at 93.5% and 13%, /θ/ at 99.25% and 3%, /q/ at 93.5% and 13%, /k/ at 97.0% and 9%, and /ʤ/ at 98.33% and 10%. This variation reflects differences in production frequency and accuracy for each sound within the sample. Additionally, the severity of errors fluctuated based on the recurrence of each sound in the dataset. Notably, /r/ is used more frequently in Arabic speech compared to /θ/, which may contribute to the observed differences in PCC-R values. Frequent exposure to /r/ may enhance articulation accuracy, whereas less commonly used phonemes like /θ/ might exhibit higher PCC-R scores due to their lower occurrence and reduced likelihood of errors.

Performance differences between individuals with concomitant misarticulations and APD and those with single misarticulations were significant for the GIN test ($Z = -3.414$, $p < 0.05$); the PPS test for right ears ($Z = -4.780$, $p < 0.05$); the PPS test for left ears ($Z = -4.224$, $p < 0.05$); the DPS test for right ears ($Z = -4.433$, $p < 0.05$); the DPS test for left ears ($Z = -4.366$, $p < 0.05$); and the RGDT ($Z = -3.981$, $p < 0.05$). However, no statistical differences were observed between the two groups in the MLD test ($Z = -0.319$, $p > 0.05$).

### 3.2. The association between APD and sound mispronunciations

Based on the classification of sound misarticulations, the number of abnormal cases was higher than normal cases for /r/ (16 cases, 43.2%), /s/ (5 cases, 13.5%), /q/ (3 cases, 8.1%), /k/ (4 cases, 10.8%), /θ/ (4 cases, 10.8%), and /ʤ/ (5 cases,

**Table 1. Descriptive Statistics of Demographics and baseline characteristics of the study participants (N = 51).**

| Variable* | Categories of Variable | N (%) |
|---|---|---|
| Gender | Male | 36 (70.6) |
| | Female | 15 (29.4) |
| Age | 6-12 Years | 45 (88.2) |
| | 13-18 Years | 6 (11.8) |
| Educational Level | Elementary (6–10 years) | 37 (72.5) |
| | Middle (11–14 years) | 8 (15.7) |
| | High (15–18 years) | 6 (11.8) |

*The variables are categorical and are described using frequencies and percentages.

13.5%). However, no statistically significant association was found between APD and sound misarticulations ($p = 0.46$, Fisher's exact test $= 4.58$; Table 2).

**3.2.1. Associations between /r/, /s/, /k/, /θ/, /dʒ/, and /q/ sound misarticulations and temporal resolution deficits as measured by the GIN test.** Before the APD test, all participants received thorough instructions, repeated up to three times if needed to ensure understanding. Additionally, screening assessments were conducted for each participant before proceeding to the main tests. Among participants with /s/ sound misarticulations, five passed the GIN test, while the remaining five did not meet the passing criteria. For the /q/ sound, one participant successfully passed the GIN test, suggesting that central auditory processing abilities significantly influence /q/ pronunciation. Four participants with /θ/ sound misarticulations were tested; after receiving instructions, two successfully passed the GIN test. Finally, five participants with /k/ sound misarticulations were examined, and three successfully passed the GIN test. However, no statistically significant association was found between temporal resolution deficits, as measured by the GIN test [20], and sound misarticulations ($p = 0.76$, Fisher's exact test $= 2.99$), as depicted in Table 3.

**3.2.2. Associations between /r/, /s/, /k/, /θ/, /dʒ/, and /q/ sound misarticulations and abnormal binaural temporal information processing as measured by the MLD test.** Of the 10 participants with /s/ sound misarticulations who took the MLD test, 26.5% failed. For the phoneme /r/, a majority (52.9%) passed, underscoring the significant role of central auditory processing in pronouncing /r/. Two participants were tested for /θ/ misarticulation, both achieving a success rate of 11.8%. Participants who took the MLD test for the /dʒ/, /q/, and /k/ phonemes also obtained successful results, with pass rates of 11.8%, 5.9%, and 8.8%, respectively. However, no statistically significant association was found between

**Table 2. Frequency distributions and percentages of APD test and sound mispronunciations in the normal group and abnormal group.**

| Sound misarticulation Groups | APD | | | | Fisher's Exact Value | p* |
| | Normal | | Abnormal | | | |
| | N | % | N | % | | |
|---|---|---|---|---|---|---|
| /r/ | 7 | 50 | 16 | 43.2 | 4.58 | 0.46 |
| /s/ | 5 | 35.7 | 5 | 13.5 | | |
| /k/ | 1 | 7.1 | 4 | 10.8 | | |
| /θ/ | 0 | 0 | 4 | 10.8 | | |
| /dʒ/ | 1 | 7.1 | 5 | 13.5 | | |
| /q/ | 0 | 0 | 3 | 8.1 | | |

N: Number of Subjects; %: Percentage of subjects; * A *p*-value below 0.05 indicates significance.

**Table 3. Frequency distributions and percentages of GIN test and sound mispronunciations in the normal group and abnormal group.**

| Sound misarticulation Groups | Temporal resolution deficits as measured by the GIN test | | | | Fisher's Exact Value | p* |
| | Normal (N = 30) | | Abnormal (N = 21) | | | |
| | N | % | N | % | 2.99 | 0.76 |
|---|---|---|---|---|---|---|
| /r/ | 16 | 53.3 | 7 | 33.3 | | |
| /s/ | 5 | 16.7 | 5 | 19.6 | | |
| /k/ | 3 | 10 | 2 | 9.5 | | |
| /θ/ | 2 | 6.7 | 2 | 9.5 | | |
| /dʒ/ | 3 | 10 | 3 | 14.3 | | |
| /q/ | 1 | 3.3 | 2 | 9.5 | | |

N: Number of Subjects; %: Percentage of subjects; *The statistical significance level was set at 0.05.

abnormal binaural interaction, as measured by the MLD test [27], and sound misarticulations ($p = 0.60$, Fisher's exact test = 3.95), as depicted in Table 4.

### 3.3. Associations between /r/, /s/, /k/, /θ/, /ʤ/, and /q/ sound misarticulations and failure to detect the minor time interval between two stimuli as measured by the RGDT

The RGDT results indicated distinct patterns. Among participants with /s/ sound misarticulations, 28.6% passed the RGDT, while 13.3% exhibited challenges indicative of auditory processing deficits. For the /r/ sound, 61.9% of participants passed the test, while 33.3% encountered difficulties. In the /k/ sound group, 4.8% of participants succeeded. Participants with /ʤ/ sound misarticulations demonstrated better performance, with 4.8% passing the RGDT and the remaining 16.7% achieving scores near the passing threshold. The /q/ and /θ/ sound groups had a 100% failure rate. A statistically significant association was found between failure to detect minor time intervals between two stimuli, as measured by the RGDT [23], and sound misarticulations ($p = 0.040$, Fisher's exact test = 9.85), as depicted in Table 5.

### 3.4. Associations between /r/, /s/, /k/, /θ/, /ʤ/, and /q/ sound misarticulations and spectral analysis difficulties as measured by the PPS Test

A total of 41 participants completed the PPS test. Of these, 17 subjects had /r/ misarticulations, 10 had /s/ sound misarticulations, 4 had /k/ misarticulations, 3 had /q/ misarticulations, 3 had /θ/ misarticulations, and 4 had /ʤ /misarticulations.

**Table 4.** Frequency distributions and percentages of MLD test and sound mispronunciations in the normal group and abnormal group.

| | Abnormal binaural temporal information processing as measured by the MLD test | | | | Fisher's Exact Value | p* |
|---|---|---|---|---|---|---|
| | Normal (N = 17) | | Abnormal (N = 34) | | | |
| Sound misarticulation Groups | N | % | N | % | | |
| /r/ | 9 | 52.9 | 14 | 41.2 | 3.95 | 0.60 |
| /s/ | 1 | 5.9 | 9 | 26.5 | | |
| /k/ | 2 | 11.8 | 3 | 8.8 | | |
| /θ/ | 2 | 11.8 | 2 | 5.9 | | |
| /ʤ/ | 2 | 11.8 | 4 | 11.8 | | |
| /q/ | 1 | 5.9 | 2 | 5.9 | | |

N: Number of Subjects; %: Percentage of subjects; *The statistical significance level was set at 0.05.

**Table 5.** Frequency distributions and percentages of RGDT test and sound mispronunciations in the normal group and abnormal group.

| | Temporal resolution deficits as measured by the RGDT test | | | | Fisher's Exact Value | p* |
|---|---|---|---|---|---|---|
| | Normal (N = 21) | | Abnormal (N = 30) | | | |
| Sound misarticulation Groups | N | % | N | % | | |
| /r/ | 13 | 61.9% | 10 | 33.3% | 9.85 | 0.04 |
| /s/ | 6 | 28.6% | 4 | 13.3% | | |
| /k/ | 1 | 4.8% | 4 | 13.3% | | |
| /θ/ | 0 | 0% | 4 | 13.3% | | |
| /ʤ/ | 1 | 4.8% | 5 | 16.7% | | |
| /q/ | 0 | 0% | 3 | 10% | | |

N: Number of Subjects; %: Percentage of subjects; *The statistical significance level was set at 0.05.

The remaining 11 participants did not complete the test because they were below the minimum age requirement of seven years.

Besides differing abnormality percentages for each sound, some participants exhibited normal results in one ear and abnormal results in the other, suggesting possible asymmetry in sound perception and processing between hemispheres. Among 17 participants with /r/ sound misarticulation, 6 indicated normal results, while the abnormality rate was 64.7% (11 out of 17). Additionally, 17.6% (3 out of 17) had normal results in one ear and abnormal results in the other. For /s/ sound misarticulation, 7 out of 10 participants obtained normal results, yielding the lowest abnormality rate of 30% (3 out of 10). Moreover, 66.6% (2 out of 3) of these participants indicated normal results in one ear and abnormal results in the other. Among participants with /k/ sound misarticulation, 1 out of 4 achieved normal results; the abnormality rate was 50% (2 out of 4), while 25% (1 out of 4) had normal results in one ear and abnormal results in the other. For /q/ sound misarticulation, 2 out of 3 participants indicated normal results, and 1 had an abnormal result, with an abnormality rate of 33.3%. Among participants with /θ/ sound misarticulation, 1 out of 3 indicated normal results, while the abnormality rate reached 66.6% (2 out of 3). Participants with /dʒ/ sound misarticulation had an abnormality rate of 50% (2 out of 4), with the remaining 2 participants indicating normal results. Deviations in the production of /q/, /θ/, and /dʒ/ sounds were not linked to differences between ears.

A one-way repeated measures ANOVA with the factor Ear revealed no significant differences in temporal ordering difficulties between both the right and left ear, as measured by the PPS test [28], attributable to sound misperception ($F_{(1,45)}$ = 0.96; $p > 0.05$; Table 6).

A linear regression on the factor Ear revealed no statistically significant relationship between sound misarticulations and spectral analysis difficulties as measured by the PPS test in the left ear ($R^2 = 0.03$, $p > 0.05$) and no statistically significant relationship in the right ear ($R^2 = 0.005$, $p > 0.05$; Fig 1).

### 3.5. Associations between /r/, /s/, /k/, /θ/, /dʒ/, and /q/ sound misarticulations and impaired pattern recognition as measured by the DPS Test

A total of 41 subjects completed the DPS test; the remaining 11 were younger than seven years. Among participants with /r/ sound misarticulation, 9 out of 17 obtained normal results, yielding an abnormality rate of 47.1%. For /s/ sound misarticulation, 8 out of 10 participants indicated normal results, with an abnormality rate of 20%. One of 3 participants with /θ/ sound misarticulation scored normal results, with an abnormality rate of 66.6%. Two out of 4 participants with /k/ and /dʒ/ sound misarticulations received normal results, yielding an abnormality percentage rate of 50%. No participants with /q/ sound misarticulation achieved normal results, with an abnormality rate of 100% (3 out of 3). The only subject with asymmetrical results had /r/ sound misarticulation, accounting for 5.9%.

A one-way repeated measures on ANOVA with the factor Ear revealed no significant differences in impaired pattern recognition between the right and left ear, as measured by the DPS test [21], attributable to sound misperception ($F_{(1,45)}$ = 1.66; $p > 0.05$; Table 7). This result indicates that impaired pattern recognition was present in both ears.

Table 6. Comparison between right ear (RE) and left ear (LE) with spectral analysis difficulties as measured by the PPS Test.

| | Descriptive Statistics | | | | | | F. value | p* |
|---|---|---|---|---|---|---|---|---|
| | Sound misarticulation classifications | | | | | | | |
| | /r/ | /s/ | /k/ | /θ/ | /dʒ/ | /q/ | | |
| Spectral analysis difficulties | | | | | | | | |
| PPS. RE% | 48.1±29.2 | 66.1±36.3 | 48.6±25.6 | 53.3±38.5 | 39.1±28.3 | 27.8±10.7 | 0.96 | 0.45 |
| PPS LE % | 52.9±31.2 | 61.1±37.2 | 44.6±18.0 | 48.33±36.7 | 41.4±30.5 | 25.5±20.1 | | |

Mean± standard error (SE). The associations between sound misarticulations and spectral analysis difficulties were measured using the PPS test; *The significance level was set at 0.05.

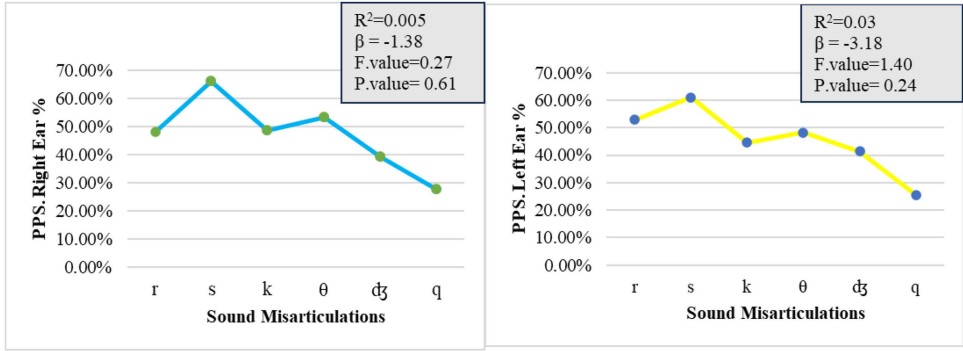

**Fig 1. Relationship between sound misarticulations and the PPS test.** Line graphs show the PPS performance for right (blue) and left (yellow) ears across misarticulated sounds (/r/, /s/, /k/, /θ/, /dʒ/, /q/). Both ears show a decline from/s/ to/q/. Regression analyses revealed weak, non-significant correlations (Right ear: R²=0.005, p=0.61; Left ear: R²=0.03, p=0.24). Significance level set at 0.05.

**Table 7. Comparison between the right ear (RE) and left ear (LE) with impaired pattern recognition as measured by the DPS Test.**

| | Descriptive Statistics | | | | | | F. value | *p** |
|---|---|---|---|---|---|---|---|---|
| | Sound misarticulation classifications | | | | | | | |
| | /r/ | /s/ | /k/ | /θ/ | /dʒ/ | /q/ | | |
| **Spectral analysis difficulties** | | | | | | | | |
| DPS. RE % | 69.1±7.3 | 80.10±11.3 | 63.3±17.2 | 42.4±18.4 | 54.1±17.6 | 18.8±9.7 | 1.66 | 0.16 |
| DPS. LE% | 66.3±6.9 | 76.6±12.3 | 62.3±18.8 | 49.9±5.4 | 52.3±18.4 | 14.4±8.0 | | |

Mean±Standard Error. The associations between sound misarticulations and impaired pattern recognition were measured using the DPS test. * The significance level was set at 0.05.

Linear regressions between sound misarticulations and spectral analysis difficulties were computed separately for each ear. The results indicated a statistically significant effect of sound misarticulations on spectral analysis difficulties in the right ear, as measured by the DPS test [21], ($R^2 = 0.09$, $p < 0.05$), and no statistically significant effect in the left ear ($R^2 = 0.06$, $p > 0.05$; Fig 2).

Participants' variables, such as age, gender, and educational level, as shown in Table 1, were considered when conducting the APD test battery. Despite the fact that the auditory system maturation varies with age, the analysis focused on linking specific sound misarticulations with APD. All participants who completed the APD test battery were evaluated using the same passing criteria, regardless of the number or severity of speech sound errors or age differences, provided they were within the target age range for the tests.

## 4. Discussion

This study aimed to examine the association between APD and specific sound misarticulations to test the hypothesis that children with persistent speech-sound misarticulations may struggle to acquire certain sounds in the presence of APD. The primary findings indicated that misarticulation may result from the central auditory system's inability to process the silent intervals of certain consonants and pitch transitions between sounds.

Linear regression analysis of the test results revealed a statistically significant relationship between sound misarticulations and three of the five tests (RGDT, PPS in the right ear only, and DPS in the right ear only), each assessing a specific auditory skill. Subjects who did not meet the passing criteria for a given test were considered to lack proficiency in the corresponding auditory skill. The presence of APD in participants with sound misarticulations was found to affect sound perception and extend the duration of the condition.

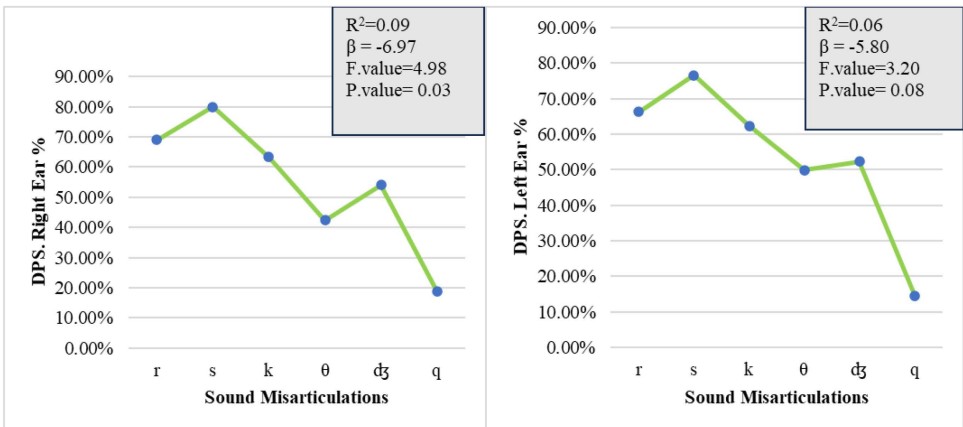

**Fig 2. Relationship between sound misarticulations and the DPS test.** Line graphs depict the percentage of correct DPS responses in the right and left ears across six sound misarticulations ( /r/, /s/, /k/, /θ/, /ð/, /q/). Right ear performance peaks at/s/ (~80%) and is lowest at/q/ (~20%), showing a significant correlation (R²=0.09, β=−6.97, F=4.98, p=0.03). Left ear performance also peaks at/s/ (~75%) and declines to/q/ (~15%), indicating a non-significant correlation (R²=0.06, β=−5.80, F=3.20, p=0.08). Significance set at 0.05.

## 4.1. The association between APD and sound mispronunciations

When the auditory system fails to accurately discriminate and process acoustic characteristics, it may lead to sound distortions in children [29]. At very rapid speech rates, auditory discrimination deteriorates further [30]. As a result, the child may struggle to accurately perceive certain acoustic features of sounds, which can negatively affect their ability to produce those sounds correctly [30]. Since the auditory cortex is considered tonotopic [31], it is suspected that speech sound frequencies in participants with abnormal auditory processing may not be sufficiently stimulated within the auditory cortex. This insufficiency can lead to poor sound discrimination and temporal resolution in the brain and an inability to detect changes (e.g., duration or silence) in auditory stimuli, ultimately resulting in sound misarticulations.

Several studies have examined the correlation between APD and various developmental and acquired disorders, such as learning disabilities, attention deficit hyperactivity disorder, and language delay. A previous study of 27 participants investigated the influence of APD on the severity of SSD in children aged 7–9 years and 11 months. The assessments included phonology, speech consistency, and metalinguistic awareness, and participants were divided into two groups based on their auditory processing test results. The study found that children with SSD, including articulation difficulties, performed poorly on auditory processing tasks. These findings align with the present study, highlighting a potential link between articulation deficits and auditory processing skills. Moreover, the severity of the articulation disorder may influence the extent of auditory processing difficulties, suggesting that more pronounced speech production issues could be associated with greater deficits in auditory discrimination, temporal processing, and auditory memory. However, it is important to distinguish between APD and articulation disorders: APD involves deficits in the brain's interpretation of auditory information, while articulation disorders are characterized by difficulties in the physical production of speech sounds. Although these conditions may coexist, they represent distinct diagnostic entities requiring separate evaluation and intervention approaches [18].

Another study involving 90 preschool children aged 6–7 years old diagnosed with specific language impairments aimed to use the APD test battery to diagnose APD in these children [29]. The study employed a battery of three binaural auditory tests developed by researchers based on behavioral audiometry. During testing, the children listened to two-syllable words presented simultaneously but separately. Children with specific language impairments had difficulty forming simple sentences using the words they heard, indicating deficits in short-term memory that contributed to language impairments.

This finding highlights the correlation between speech and language impairments and APD, consistent with the results of the present study [32].

In 2014, a study at Alexandria University Children's Hospital investigated the central auditory pathways and the correlation between APD and language delay in children diagnosed with autism spectrum disorder (ASD). The study included 30 children diagnosed with ASD according to the Diagnostic and Statistical Manual of Mental Disorders (fourth edition, text revision) and the Autism Diagnostic Interview-Revised. Behavioral and electrophysiological tests were used to diagnose APD and assess the response of the central auditory system to various acoustic stimuli. The children's ages were considered, as behavioral tests were unsuitable for those under seven years old; thus, age-appropriate assessments were selected based on the neural maturation of their auditory systems. The study confirmed a correlation between APD and limited language development in children with ASD, implicating APD as a contributing pathology of ASD. It also highlighted that children with ASD exhibit an immature central auditory system at both cortical and brainstem levels [33].

Another study of 34 participants investigated the associations between APD, language, and cognition in children [4]. Participants underwent an APD test battery comprising the Frequency Patterns, Dichotic Digits, and Competing Sentences tests. Scores on the APD battery were correlated with results from the Clinical Evaluation of Language Fundamentals and the Wechsler Intelligence Scale for Children. Central auditory test results indicated significant correlations with intelligence quotient and working memory but no correlation with language assessments, and the reasons for these results were unclear. Additionally, Dichotic Digits and Frequency Patterns test results correlated with cognition test outcomes, particularly in children with mild to moderate Clinical Evaluation of Language Fundamentals scores and borderline low Wechsler Intelligence Scale for Children scores. In contrast, Competing Sentence scores indicated no correlation with cognition or language assessments. The study also highlighted that language and cognition measures were more strongly correlated with each other than with APD.

Many studies have stated findings consistent with the present research. In 2023, a systematic review of six articles examined the effect of APD on speech development in children with SSD [34]. The review explored how APD may contribute to speech disorders at the levels of semantics, syntax, phonology, and sound acquisition. It demonstrated that APD affects temporal processing abilities and phonological awareness, thereby impacting language development [34].

A study by [35] investigated the influence of emotion in subjects with normal central auditory processing to enhance speech therapy for children with stuttering and misarticulations. The study involved 80 subjects who underwent auditory and speech evaluations to confirm the presence of a speech disorder. The findings confirmed the effectiveness of incorporating emotional engagement and family involvement to improve speech therapy outcomes. Subjects with stuttering exhibited fewer disfluencies when hearing familiar voices, and those with sound misarticulations demonstrated higher correct pronunciation rates with familiar voices compared to unfamiliar ones.

## 4.2.  Phonological errors in speech production exhibit a correlation with specific central auditory processing outcomes

During the early stages of language acquisition, presenting phonemic sequences at a reduced rate facilitates both recognition and reproduction. This phenomenon suggests that as individuals become more adept at interpreting temporal variations, their ability to process auditory information may improve [30,36]. The diagnosis of APD, conducted by audiologists, requires precision and care. A multidisciplinary team approach is preferred for managing APD, with treatment tailored to the individual. This contrasts with phonological disorders, where speech therapists primarily focus on correcting sound errors and generalizing improvements [3].

Analysis of our study data indicated that participants with misarticulations in /s/ and /r/ sounds exhibited the highest rates of abnormal central auditory processing results. This may be because /s/ and /r/ are acquired in the final stages of phonological development [37]. Additionally, regardless of the type of sound misarticulation, most participants performed worse on the MLD test compared to other components of the auditory processing battery. Lower MLD scores may indicate

deficits in brainstem pathways involved in binaural processing, underscoring that the ability to process competing auditory signals is a critical auditory skill for speech development [38].

Furthermore, our investigation indicated that participants with /r/ and /dʒ/ misarticulations performed poorly on the PPS test, indicating challenges in understanding complex auditory information, pattern recognition, and temporal processing [19,28]. The /r/ sound contains many harmonics, contributing to its distinctive acoustic properties. Its quality is shaped by the involvement of the tongue, lips, and jaw, with tongue position varying during production, creating multiple resonances in the vocal tract. Thus, children with /r/ misarticulation are likely to struggle with analyzing the simultaneous presence of different pitches [39]. Additionally, /dʒ/ is an affricate that combines features of both stop and fricative sounds; it begins with a complete closure of the vocal tract (similar to a stop) and is then released, creating friction (similar to a fricative). To acquire production of a complex sound like /dʒ/, children must attend closely to formant variations [40].

Participants with /s/ misarticulation demonstrated better results on auditory processing tests compared to those with other misarticulations. Nevertheless, they faced challenges on the MLD and RGDT, as they could only perceive the masking noise, not the tone. Background noise can obscure subtle acoustic cues, including the hissing quality of /s/, which can impair speech clarity and hinder learning to pronounce /s/ in children who cannot effectively filter speech from noise. The RGDT is particularly affected by /s/ misarticulation because sibilant sounds like /s/ require complex temporal processing involving multiple phases: sibilant onset, sustained duration, release, and transition [41]. The MLD test yielded the poorest performance among auditory processing subtests for participants with abnormal /θ/ production, likely because this high-pitched consonant's audible friction can be masked by background noise [42]. Finally, subjects with /q/ misarticulation exhibited the poorest scores on the DPS test, suggesting that stop sounds like /q/, which have rapid and dynamic temporal characteristics, require precise detection of fast-changing production patterns [43]. Therefore, incorrect pronunciation of the /q/ sound may be linked to abnormal DPS test results.

Two issues may undermine the robustness of the current study's methodology and its replicability. First, some participants had difficulty completing the auditory processing subtests due to an inadequate understanding of the test instructions. Second, nonlinguistic tests were used to evaluate auditory processing because linguistically tailored tests specifically designed for Jordanian speakers were unavailable.

## 5. Conclusion

This study explored the potential co-occurrence of articulation disorders and APD. Children with certain specific sound misarticulations appear more susceptible to APD than others. Moreover, each sound demonstrated suboptimal performance on distinct auditory processing tests, indicating specific deficits in auditory skills. The findings suggest that participants diagnosed with both sound misarticulations and APD may require an extended duration of traditional speech therapy. It is also recommended that they undergo APD therapy to target the specific auditory skills they struggle with, such as auditory discrimination, ordering, and sequencing. Audiologists and speech therapists should collaborate in diagnosing and managing these disorders. Future research should observe the prognosis of children with misarticulations while focusing on specific auditory processing skills likely to impact the production of target sounds.

## Supporting information

**S1 Data. Dataset- unknown.**
(XLSX)

## Author contributions

**Conceptualization:** Safa Alqudah.

**Data curation:** Safa Alqudah.

**Formal analysis:** Safa Alqudah.

**Funding acquisition:** Safa Alqudah.

**Investigation:** Safa Alqudah.

**Methodology:** Lama Al-rashed, Lelas Mansi.

**Project administration:** Safa Alqudah.

**Resources:** Safa Alqudah.

**Software:** Safa Alqudah.

**Supervision:** Safa Alqudah.

**Validation:** Safa Alqudah.

**Visualization:** Safa Alqudah.

**Writing – original draft:** Safa Alqudah, Lama Al-rashed, Lelas Mansi.

**Writing – review & editing:** Safa Alqudah, Lama Al-rashed.

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
