## [Decision Letter · Decision Letter 0]

12 Feb 2025

PONE-D-24-46256THE RELATIONSHIP BETWEEN CENTRAL AUDITORY PROCESSING DISORDER AND ARTICULATION DISORDERSPLOS ONE

Dear Dr. Alqudah,

Thank you for submitting your manuscript to PLOS ONE. After careful consideration, we feel that it has merit but does not fully meet PLOS ONE’s publication criteria as it currently stands. Therefore, we invite you to submit a revised version of the manuscript that addresses the points raised during the review process.

Specifically, please address the raised issues of the reviewers in a point by point response and make the corresponding changes in the manuscript, highlighting the new passages.

We look forward to receiving your revised manuscript.

Kind regards,

Andreas Buechner

Academic Editor

PLOS ONE

Reviewers' comments:

Reviewer's Responses to Questions

**Comments to the Author**

1. Is the manuscript technically sound, and do the data support the conclusions?

Reviewer #1: Partly

Reviewer #2: Partly

2. Has the statistical analysis been performed appropriately and rigorously? 

Reviewer #1: Yes

Reviewer #2: No

3. Have the authors made all data underlying the findings in their manuscript fully available?

Reviewer #1: Yes

Reviewer #2: No

4. Is the manuscript presented in an intelligible fashion and written in standard English?

Reviewer #1: Yes

Reviewer #2: No

5. Review Comments to the Author

Reviewer #1: The strengths of this article lie in several key areas. It addresses an underexplored topic, providing new insights into how auditory processing deficits may influence specific sound misarticulations. The study builds on existing knowledge of speech and language development while addressing the limited understanding of the relationship between different auditory processing skills and articulation disorders. Methodologically, it employs a validated test protocol, ensuring a reliable assessment of auditory processing capabilities. Additionally, the authors use appropriate statistical methods, such as regression and Fisher's exact test, to analyze associations between CAPD and sound misarticulations. Finally, the study's relevance is clear, as it has direct implications for clinical practice.

The study has strengths but also several weaknesses that need attention. While the manuscript is generally well-structured, the background does not clearly explain why specific misarticulations were chosen, leading to some confusion. The reliance on non-linguistic APD tests may not fully address the linguistic challenges of the participants, which limits the clinical relevance of the findings and should be acknowledged. It is also unclear how task comprehension was assessed or addressed, which could have affected the results. Additionally, there is no discussion about whether age influenced the results. The discussion section shuld also explore why certain auditory processing skills are related to specific misarticulations. These issues should be addressed to improve the clarity and overall impact of the study.

Specific Comments

The specific comments provided are examples, but there may be similar issues elsewhere in the manuscript that are not explicitly mentioned. I encourage the authors to carefully review the entire manuscript for consistency and clarity in these areas.

Line 207 (Table 1): The article refers to participants' educational levels (elementary, middle, and high school) but does not define the corresponding age ranges, which may not be clear for international readers. This is particularly important because of the unequal distribution between groups, which could influence the interpretation of the results.

Line 201 - 206: The reported p-values (p > 0.001) appear inconsistent with the interpretation of significant differences. For the MLD test, while p < 0.01 is stated, the Z-value (-0.319) suggests minimal differences. Please ensure consistency and clarity in reporting the statistical results.

Line 213: The presentation of information is unclear. The numbers in brackets do not align properly with the corresponding sound misarticulations. Additionally, the way it is written seems to imply a significant difference exist. A more precise and structured presentation is needed to avoid confusion and accurately convey the findings.

Example Line 230 and 244 (In results section): It is unclear how it was determined that a child understood or misunderstood a task. Also, how the CAPD tests were specifically applied to evaluate misarticulations of individual sounds (Eg., “Out of the ten participants who took the MLD test for the /s/ sound, 26.5% failed”). Providing a more detailed description of the procedures in the Methods section would be helpful, or the results could be written more clearly when interpreted.

Line 249 and Line 306: The statement attributing temporal information processing to the MLD test and spectral analysis to the PPS test is inaccurate. The MLD test is primarily a measure of binaural interaction and PPS is measuring temporal sequencing. While pitch changes in PPS are related to frequency, the test does not evaluate the listener’s ability to analyze the spectral components of a sound. I suggest to include relevant citations to justify the connection.

Line 456: What does “struggle” mean in this context? Does it imply that subjects performed poorly or that they had difficulty understanding the task?

Line 461: Poor performance in PPS does not mean that poor discrimination between frequencies.

Minor Issues

1.The manuscript alternates between using "CAPD" and "APD" to refer to central auditory processing disorder. Consistently using one term throughout the text would improve coherence.

2.The way significance values are reported is inconsistent. Some are written as "p<0.05," while others are presented as "p value < 0.05."

3.The phrasing “The statistical significance level was set at α ≤ 0.05” below each Table could be misleading, as it suggests a range rather than a fixed threshold. For clarity, it would be better to state, α = 0.05.

4.Line 371: The phrase “Prolong the prognosis” is not commonly used in clinical context and its meaning is ambiguous.

Reviewer #2: The paper has several fundamental issues:

1. It does not clarify whether the study participants have been diagnosed with Speech Sound Disorder (SSD). While the authors mention that the inclusion criterion is misarticulation of specific speech sounds and refers to articulatory disorder, it remains unclear whether the participants have idiopathic SSD or SSD resulting from known causes, such as Complex Neurodevelopmental Disorders. Although the speech manifestations may appear similar, the underlying causes differ significantly.

2. A key aspect to consider is that the diagnosis of speech sound disorders (SSD) must be thorough and specify the type of disorder, irrespective of the classification system used (whether Shriberg et al. or Dodd et al.). It is essential to clarify whether the disorder is cognitive-linguistic (phonological), involves speech errors (sound distortion), or is related to speech motor. Additionally, SSD is classified as persistent if it continues beyond the age of 9. Therefore, it is important to clearly distinguish the analysis of data from children aged 6 years to 18 years.

3. When it comes to diagnosing Auditory Processing Disorder (APD), the specific literature recommends conducting tests for APD starting at the age of 7. It is important to specify the normality criteria for different age groups for each test, particularly for 6-year-olds. This information should be clearly explained, and the adopted normality criteria should be presented.

4. Page 15: Why do not the authors cite the 2007 paper, which is accessible to all readers?

5. Page 26-27, lines 388-394: Revise the reference, as the study does not mention hyperactivity.

6. In general, it is important to clarify the results. Were the analyses conducted with participants of all ages grouped together? Do 6-year-olds perform as well as 18-year-olds in auditory processing tests? At this developmental stage, does age play a role? Additionally, why were children with only one speech error not analyzed separately from those with two or more errors? Is there a correlation between the severity of errors and auditory processing disorders (APD)? Another relevant point regarding speech sounds is that when the error is phonological, it may occur as a substitution for a phonological feature or rule of the language. Therefore, it would be interesting to know if everyone who made errors in the /s/ sound made the same type of error.

7. Page 16 lines 196-200. It was not specified how many participants exhibited each type of sound error, which is crucial for understanding the results and supporting the discussion. Additionally, it is important to identify the type of error observed—whether it was a distortion or a phonological error. If it is phonological, we should note the specific phonological process involved. Furthermore, clarifying whether the errors involved the omission or substitution of sounds would be beneficial. It would also be helpful to know how many sounds each participant struggles with, as this information can aid in determining the severity of the Speech Sound Disorder (SSD). You might consider using the Percent Consonants Correct (PCC) or the PCC-R, as described by Shriberg et al., to correlate severity with Auditory Processing Disorder (APD).

8. Page 27 lines 395-396 This refers to another communication disorder with distinct characteristics and symptoms.

6. PLOS authors have the option to publish the peer review history of their article (what does this mean? ). If published, this will include your full peer review and any attached files.

**Do you want your identity to be public for this peer review?** For information about this choice, including consent withdrawal, please see our Privacy Policy .

Reviewer #1: No

Reviewer #2: No

---

## [Author Response · Author response to Decision Letter 1]

8 May 2025

The response to the reviewers has been uploaded as a file in the 'Attach Files' section.

---

## [Decision Letter · Decision Letter 1]

28 Jul 2025

PONE-D-24-46256R1THE RELATIONSHIP BETWEEN AUDITORY PROCESSING DISORDER AND ARTICULATION DISORDERSPLOS ONE

Dear Dr. Alqudah,

Thank you for submitting your manuscript to PLOS ONE. After careful consideration, we feel that it has merit but does not fully meet PLOS ONE’s publication criteria as it currently stands. Therefore, we invite you to submit a revised version of the manuscript that addresses the points raised during the review process.

We look forward to receiving your revised manuscript.

Kind regards,

Andreas Buechner

Academic Editor

PLOS ONE

Journal Requirements:

Reviewers' comments:

Reviewer's Responses to Questions

**Comments to the Author**

1. If the authors have adequately addressed your comments raised in a previous round of review and you feel that this manuscript is now acceptable for publication, you may indicate that here to bypass the “Comments to the Author” section, enter your conflict of interest statement in the “Confidential to Editor” section, and submit your "Accept" recommendation.

Reviewer #1: (No Response)

Reviewer #2: (No Response)

2. Is the manuscript technically sound, and do the data support the conclusions?

Reviewer #1: Yes

Reviewer #2: Partly

3. Has the statistical analysis been performed appropriately and rigorously? 

Reviewer #1: Yes

Reviewer #2: Yes

4. Have the authors made all data underlying the findings in their manuscript fully available?

Reviewer #1: Yes

Reviewer #2: Yes

5. Is the manuscript presented in an intelligible fashion and written in standard English?

Reviewer #1: Yes

Reviewer #2: Yes

6. Review Comments to the Author

Reviewer #1: Thank you for the thoughtful revisions addressing the prior comments. The manuscript is improved overall. However, I still recommend minor revision as stated below.

Abstract (results): The statement “Central auditory processing evaluations revealed abnormalities in 36 of 51 participants (36%)” appears inconsistent with the findings reported in the results section. Please correct or clarify.

Line 127: Typo. “complaints limited to of misarticulations (pure articulation disorders based on SLP evaluations) of the /r/..” /

Line 127: “administrated” is incorrect, should be “administered”.

Line 260: Please include “ …"binaural" temporal information processing as measured by MLD” to be more accurate.

Line 384: Sentence is inconsistent, “ As a result, the child may be able to process certain acoustic features of sounds correctly, affecting sound production.” Please revise for clarity.

Reviewer #2: The paper is much clearer after the reformulations. Despite the revision regarding the definitions and characteristics of SSD, which are now better defined, on page 16, from line 203, there is a statement that substitution and distortion of sounds are classified as articulatory. This statement undermines the entire characterization of the studied population, since substitution is a phonological error regardless of the phonological analysis model used. Only sound distortion is classified as articulatory and involves imprecision in the speed or synchronization of the articulators. Both phonological and articulatory errors are typical of children with idiopathic SSD.

The PCC-R was then calculated for each sound, which is unusual because it is a metric used for a speech sample where all the sounds of the language can occur. It should be noted that the PCC-R does not treat distortions as errors and therefore mainly indicates phonological errors. For example, the PCC-R value of 97% for the sound /ɵ/ shows that it has no phonological errors, as it is close to the 100% maximum, but we do not know the percentage of errors caused by distortion.

Clarifying that the children in the study had idiopathic SSD is sufficient, but distinguishing substitution errors from distortions would make the analyses more robust. The results presented are described in greater detail, as is the discussion. The main contribution of this paper is to highlight that, depending on the error—whether phonological or articulatory—a detailed analysis of auditory processing based on the type of error observed in the SSD can support treatment with the

7. PLOS authors have the option to publish the peer review history of their article (what does this mean? ). If published, this will include your full peer review and any attached files.

**Do you want your identity to be public for this peer review?** For information about this choice, including consent withdrawal, please see our Privacy Policy .

Reviewer #1: No

Reviewer #2: No

---

## [Author Response · Author response to Decision Letter 2]

4 Aug 2025

Please refer to the attached file in the 'Attachments' section for our detailed response to the reviewers' comments.

---

## [Decision Letter · Decision Letter 2]

8 Sep 2025

THE RELATIONSHIP BETWEEN AUDITORY PROCESSING DISORDER AND ARTICULATION DISORDERS

PONE-D-24-46256R2

Dear Dr. Alqudah,

We’re pleased to inform you that your manuscript has been judged scientifically suitable for publication and will be formally accepted for publication once it meets all outstanding technical requirements.

Kind regards,

Andreas Buechner

Academic Editor

PLOS ONE

Additional Editor Comments (optional):

Reviewer #1:

Reviewer #2:

Reviewers' comments:

Reviewer's Responses to Questions

**Comments to the Author**

1. If the authors have adequately addressed your comments raised in a previous round of review and you feel that this manuscript is now acceptable for publication, you may indicate that here to bypass the “Comments to the Author” section, enter your conflict of interest statement in the “Confidential to Editor” section, and submit your "Accept" recommendation.

Reviewer #1: All comments have been addressed

Reviewer #2: All comments have been addressed

2. Is the manuscript technically sound, and do the data support the conclusions?

Reviewer #1: Yes

Reviewer #2: Yes

3. Has the statistical analysis been performed appropriately and rigorously? 

Reviewer #1: Yes

Reviewer #2: Yes

4. Have the authors made all data underlying the findings in their manuscript fully available?

Reviewer #1: Yes

Reviewer #2: Yes

5. Is the manuscript presented in an intelligible fashion and written in standard English?

Reviewer #1: Yes

Reviewer #2: Yes

6. Review Comments to the Author

Reviewer #1: (No Response)

Reviewer #2: The authors fulfilled the requests from the previous opinion, and the developed study can be shared with other speech-language pathologists and audiologists.

7. PLOS authors have the option to publish the peer review history of their article (what does this mean? ). If published, this will include your full peer review and any attached files.

**Do you want your identity to be public for this peer review?** For information about this choice, including consent withdrawal, please see our Privacy Policy .

Reviewer #1: No

Reviewer #2: No

---

## [Editor Report · Acceptance letter]

PONE-D-24-46256R2

PLOS ONE

Dear Dr. Alqudah,

I'm pleased to inform you that your manuscript has been deemed suitable for publication in PLOS ONE. Congratulations! Your manuscript is now being handed over to our production team.

Kind regards,

on behalf of

Andreas Buechner

Academic Editor

PLOS ONE